# Vaccine hesitancy and HPV vaccine uptake among male and female youth in Switzerland: a cross-sectional study

Laura M Kiener ,[1,2] Corina L Schwendener ,[1,2] Kristen Jafflin,[2,3]
Audrey Meier,[1,2] Noah Reber,[1,2] Susanna Schärli Maurer,[4] Franco Muggli,[5]
Nejla Gültekin,[6] Benedikt M Huber,[7] Sonja Merten,[2,3] Michael J Deml ,[2,3]
Philip E Tarr [1,2]

LMK and CLS contributed equally.

For numbered affiliations see end of article.

**Correspondence to**
Professor Philip E Tarr;
philip.tarr@unibas.ch

## ABSTRACT

**Objectives** Identifying factors associated with human papillomavirus (HPV) vaccine uptake is essential for designing successful vaccination programmes. We aimed to examine the association between vaccine hesitancy (VH) and HPV vaccine uptake among male and female youth in Switzerland.

**Design** With a cross-sectional study, an interview-based questionnaire was used to collect information on sociodemographic factors, vaccination records and to measure the prevalence of VH using the Youth Attitudes about Vaccines scale (YAV-5), a modified version of the Parent Attitudes about Childhood Vaccinations survey instrument.

**Setting and participants** Eligible male and female participants, 15–26 years of age, were recruited through physicians' offices and military enlistment in all three language regions of Switzerland. Of 1001 participants, we included 674 participants with a vaccination record available (415 males and 259 females) in this study.

**Primary and secondary outcome measures** The outcome was uptake for HPV vaccine (having received ≥1 dose of HPV vaccine). Covariates were VH, sex, age and other sociodemographics.

**Results** 151 (58%) female and 64 (15%) male participants received ≥1 dose of HPV vaccine. 81 (31%) female and 92 (22%) male participants were VH (YAV-5-Score >50). The odds for being unvaccinated were higher for VH women than non-VH women, adjusted OR=4.90 (95% CI 2.53 to 9.50), but similar among VH and non-VH men, OR=1.90 (95% CI 0.84 to 4.31). The odds for being unvaccinated were lower for younger men (born on or after 1 July 2002) than older men (born before 1 July 2002), OR=0.34 (95% CI 0.14 to 0.81), but we found no association between age and vaccine uptake for female youth, OR=0.97 (95% CI 0.48 to 1.97).

**Conclusions** VH was associated with lower HPV vaccine uptake in female youth but not male youth in our study population in Switzerland. Our findings suggest that issues other than VH contribute to HPV underimmunisation in male youth in Switzerland.

## INTRODUCTION

Identifying factors contributing to under-immunisation is essential for designing

## Strengths and limitations of this study

► This study is the first using the Youth Attitudes about Vaccines scale-5 to measure vaccine hesitancy (VH) in youth and to associate VH with HPV vaccination uptake.

► We include women and a large number of male youth from all three language regions of Switzerland, which allowed us to gain important insight into men's perspectives and vaccine uptake.

► The findings of this study are likely comparable to other high-income countries, in which human papillomavirus (HPV) vaccination programmes have recently been extended to young men and where HPV vaccine is covered by health insurance or state-funded programmes.

► The large age range of the participants might diminish effects of HPV vaccine recommendations and programmes since programmes have changed frequently in the past years.

► As our participant sampling is not representative, VH participants might be less likely than non-VH persons to participate in a study examining VH.

successful human papillomavirus (HPV) vaccination programmes. Previous studies have shown various influences on HPV under-immunisation in female youth, including access barriers, lack of efficient school vaccination programmes and concerns about safety and side effects.[1–9] Because many countries have only recently recommended HPV vaccine for men, information on the determinants of HPV vaccine uptake in men is limited, with lack of awareness and knowledge, access barriers and lack of healthcare provider recommendation being previously reported determinants.[4–6 10–17]

Vaccine hesitancy (VH), which 'refers to delay in acceptance or refusal of vaccination despite availability of vaccination services'[18] was identified by the WHO as among 10 threats to global health in 2019.[19] Parental

VH contributes to underimmunisation of their children for both childhood vaccines[20–22] and HPV vaccine.[23 24]

Since the determinants of HPV vaccination are context specific, local studies are needed. Additionally, previous studies investigating VH towards HPV vaccine had a limited sample size[24] or only investigated parents' perspectives without taking into account youth perspectives.[13 17 23 25–28] Exploring parents' perspectives is appropriate in the target group for HPV vaccine, that is, adolescents 11–14 years of age, ideally before starting sexual activity. However, many countries, including Switzerland, allow adolescents as young as 14 to make vaccine decisions.[29] In addition, HPV vaccine is now recommended as a catch-up vaccine until 26 years in many countries, and until age 45 in the USA.[30] Therefore, youth and young adult attitudes and perspectives on HPV vaccine, including VH, merit further investigation on a larger scale. This study addresses these research gaps by examining youth's VH in relationship to their HPV vaccination uptake.

The Swiss Federal Office of Public Health (FOPH) and the Federal Vaccination Commission have recommended HPV vaccine since 2007 as a routine vaccine for girls 11–14 years of age and young women as catch-up vaccination 15–26 years of age.[31] In 2015, the FOPH added HPV vaccine to the list of supplementary vaccines for boys and young men with the same age groups as for women.[32] HPV vaccine is a key vaccine included in the Swiss National Vaccination Strategy,[33] and cantonal (state) vaccination programmes fully cover the cost of HPV vaccination for both men and women in the recommended age groups. Nevertheless, HPV vaccination coverage in Switzerland remains low.[34 35] While vaccine uptake has increased in the last decade,[2] in the most recent assessment by the FOPH (period 2017–2019), only 64% and 20% of 16-year old girls and boys, respectively, had received ≥1 dose of HPV vaccine.[34] Regional differences in uptake have been attributed to sociocultural factors, differences in vaccination policies and differences in the strategies and efforts of the local health departments.[36 37]

We use the Youth Attitudes about Vaccines Scale (YAV-5), a modified version of the Parent Attitudes about Childhood Vaccination (PACV) tool, originally developed by Opel and colleagues, in order to measure youth VH.[38] The PACV is a validated measure of VH predicting underimmunisation in children in the USA.[20] However, researchers found no association between PACV score and underimmunisation with adolescent vaccines, including HPV, in the USA.[39] With the later introduction of the HPV vaccine recommendation for men, we expect factors other than VH, such as HPV vaccine awareness, to influence uptake in men. The objectives of this study are to particularly assess how VH influences HPV vaccine uptake among men and women 15–26 years old in our sample in Switzerland. We hypothesise that (1) a YAV-5-score >50 (indicating VH) in youth is associated with lower HPV vaccine uptake and (2) the association between YAV-5-score and HPV vaccine uptake is stronger among women than among men.

## METHODS

We conducted this cross-sectional study in the context of our Swiss National Research Program NRP74 on the determinants of VH and underimmunisation with HPV and childhood vaccines in Switzerland.[40] All participants provided informed consent after the nature and possible consequences of the study had been fully explained. Full details on the study, including recruitment methods, power calculation, development of the survey and a description of the situation of biomedicine and complementary and alternative medicine (CAM) in Switzerland have been previously published.[41]

### Sampling

Participants were male and female youth 15–26 years of age. We recruited participants in the offices of physicians providing either conventional biomedicine alone ('biomedical provider') or biomedicine combined with complementary and alternative medicine ('CAM provider'. We recruited participants from January 2019 to May 2020 in urban and rural areas in the three main language regions of Switzerland (German, French and Italian). In order to supplement the low number of male participants, we also recruited study participants during their military enlistment in the Swiss Army. Military service is mandatory for all Swiss men, with enlistment taking place at around 19 years of age. Participants recruited in physicians' offices were interviewed on the phone after the physician visit. Participants recruited through the military were interviewed face-to-face on site. Medical students trained by senior researchers in recruiting and obtaining informed consent conducted the interviews, which lasted 25–35 min. Data collectors entered all data into open data kit using tablets.

### YAV-5

We measured VH using the YAV-5, an adapted version of the PACV survey instrument, which was originally developed by Opel and colleagues.[38] We have previously validated a 5-item short version of the PACV in German, French, Italian and English regarding childhood vaccines.[42] For the present project, we adapted the questions for use in youth and validated the YAV-5 in German, French, Italian and English language.[43] The adaptions we made for the YAV-5 instrument involved asking youth about their attitudes about adolescent vaccines. Each YAV-5-item is scored from 0 to 2. Individual scores are then summed and transformed to a 0–100 scale using simple linear transformation.[20 44] We will refer to participants with a score <50 as non-VH and participants with a score ≥50 as VH.[20 45]

### Survey

In addition to the YAV-5, the survey included items about participants' sociodemographic characteristics such as language and birthdate. After finalising the survey in English, we translated it into German, French and Italian using the forward and backward method,[41] pretested,

adjusted and piloted the survey as previously described.[42] We determine youths' HPV vaccination status through an examination of copies of their vaccination record and, for those not having a vaccination record because they have not received any vaccines, a personal statement to this effect. Being unvaccinated was defined as not having received ≥1 dose of HPV vaccine at the time of the interview.

## Participant age groups

We divided participants into two age groups based on the later introduction of the recommendation of the HPV vaccine for boys in Switzerland. HPV vaccine has been covered by health insurance for male adolescents 11–14 years of age and as a catch-up vaccination until the age of 26 years since 1 July 2016. We, therefore, refer to male participants who were 14 years of age or older when insurance coverage for HPV vaccination began (born before 1 July 2002) as 'older' and those who were under 14 at this time (born on or after 1 July 2002) as 'younger' participants, respectively. For comparison, we applied the same age cut-offs to female participants.

## Patient and public involvement

This study did not include patient or public involvement in designing the study, commenting the outcomes, interpreting the results or reviewing this manuscript.

## Quantitative data analysis

We present descriptive statistics to summarise participants' age, place of birth, language region, YAV-5 score and their HPV vaccination status. We calculated unadjusted ORs for being unvaccinated for all independent variables. We then conducted multilevel logistic regressions with random effects by recruitment setting (military enlistment, biomedical office, CAM office) to calculate adjusted ORs. Due to the high prevalence of VH, calculated ORs are significantly higher than prevalence ratios would be, however, as these models are standard in the literature and allow easier adjustment for sampling frames, we use them here. Control variables included younger versus older age group, being Swiss born, language region and living in an area with local HPV vaccine school programmes (model 0). Next, we extended the model by adding the YAV-5-score as a new control variable for testing our hypotheses (model 1). We also stratified analyses by sex. Additionally, we also checked for an interaction between VH and age, but found none. Results of those analyses are available on request. We analysed all data using STATA (V.12.0, Stata Corp, College Station, Texas).

## RESULTS
## Participants

We completed interviews with 1001 youth. For our analysis, we exclude youth who did not match our age criteria (n=15), who did not provide a correct postcode (n=10) or who did not provide vaccination records for any reason other than not having one due to not vaccinating (n=302).

All analyses are, therefore, based on 674 participants, of which 415 (62%) are men and 259 (38%) are women. Their characteristics are shown in table 1. The majority of participants were Swiss born and lived in the German-speaking region of Switzerland, consistent with this being the largest Swiss language region. We recruited the majority of male participants during military enlistment and the majority of female participants in biomedical physicians' offices. We recruited more female than male participants in CAM physicians' offices. 151 (58%) female and 64 (15%) male participants had received ≥1 dose of HPV vaccine. 81 (31%) females and 92 (22%) males were VH with a YAV-5-score ≥50.

## Determinants of HPV vaccine uptake, entire study population

In univariable analysis, the odds of being unvaccinated with HPV were higher for men than for women (OR=5.28; 95% CI 3.17 to 8.80) and higher among participants recruited during military enlistment, most of whom were male, and in CAM offices (table 2). VH youth had higher odds of being unvaccinated than non-VH youth, OR=2.75 (95% CI 1.69 to 4.48). We found no significant difference in the odds of being unvaccinated in the univariable analysis based on age, being Swiss-born, language region or living in areas with local school vaccination programmes. In the multivariable model 0, the odds were similar to the odds calculated in univariable analysis, with the exception of Swiss-born participants, whose odds for being unvaccinated were higher than foreign-born participants (table 2). In the multivariable model 1, after adding VH, the odds remained essentially unchanged: as in univariable analysis, the odds of being unvaccinated were higher among VH participants than non-VH participants, OR=3.20 (95% CI 1.94 to 5.26).

## Determinants of HPV vaccine uptake, stratified by sex

The univariable odds for being unvaccinated were lower for younger men compared with older men, but we found no significant difference in the odds among younger and older women (table 3). The univariable odds for being unvaccinated were higher for women recruited in CAM offices than for women recruited in biomedical offices, and the odds for being unvaccinated were higher in VH women than non-VH women, OR=5.29 (95% CI 2.71 to 10.33). We found no correlation between recruitment setting or VH and vaccine uptake for men (table 3). There was no significant association between place of birth, language region and school vaccination programmes and vaccine uptake. In the multivariable model 0 and model 1, results for both males and females were similar to univariable results (table 3). The odds for being unvaccinated were higher among VH women than non-VH women, OR=4.90 (95% CI 2.53 to 9.50), but we found no association between VH and vaccine uptake for men, OR=1.90 (95% CI 0.84 to 4.31). Similarly, the odds for being unvaccinated were higher among women recruited

**Table 1** Participant characteristics

| | Total (n=674) | Male (n=415) | Female (n=259) |
|---|---|---|---|
| | n (%) | n (%) | n (%) |
| Age in years (median, IQR) | 19.1 (18.1, 20.7) | 19.2 (18.4, 20.1) | 19.0 (16.8, 21.7) |
| Born before 1 July 2002 | 563 (83.5) | 370 (89.2) | 193 (74.5) |
| Born on or after 1 July 2002 | 111 (16.5) | 45 (10.8) | 66 (25.5) |
| Swissborn | 621 (92.1) | 382 (92.1) | 239 (92.3) |
| Language region | | | |
| German | 494 (73.3) | 321 (77.4) | 173 (66.8) |
| French | 43 (6.4) | 16 (3.9) | 27 (10.4) |
| Italian | 137 (20.3) | 78 (18.8) | 59 (22.8) |
| Participant recruitment setting | | | |
| Military enlistment | 287 (42.6) | 283 (68.2) | 4 (1.5) |
| Biomedical provider | 280 (41.5) | 97 (23.4) | 183 (70.7) |
| CAM provider | 107 (15.9) | 35 (8.4) | 72 (27.8) |
| HPV vaccine school programme in area of their residence | | | |
| School programme available | 366 (54.3) | 232 (55.9) | 134 (51.7) |
| Vaccination status | | | |
| Has received ≥1 dose of HPV vaccine | 215 (31.9) | 64 (15.4) | 151 (58.3) |
| YAV-5-Score | | | |
| Score ≥50 (vaccine hesitant) | 173 (25.7) | 92 (22.2) | 81 (31.3) |
| YAV-5-Score mean (SD) | 32.3 (23.4) | 30.4 (22.3) | 35.3 (24.8) |
| YAV-5-Score median (IQR) | 30.0 (20, 50) | 30.0 (10, 40) | 30.0 (20, 50) |

CAM, complementary and alternative medicine; HPV, human papillomavirus; IQR, interquartile range; SD, standard deviation; YAV-5, Youth Attitudes about Vaccines Scale.

in a CAM office than women recruited in a biomedical office even when controlling for VH in model 1, OR=2.52 (95% CI 1.13 to 5.63), but for men, we found no association between recruitment setting and vaccine uptake, OR=1.43 (95% CI 0.39 to 5.20).

## DISCUSSION

Here we show that male youth have higher odds of being unvaccinated against HPV than women, despite fewer men being VH in our study. We found a significant association between VH and HPV underimmunisation only for women. The main finding of the present study shows that VH is a major determinant of HPV underimmunisation for female youth but not for male youth.

HPV vaccine uptake in our study was far lower in men than in women, despite the HPV vaccine recommendation having been officially made >4 years prior to the interviews. Our finding of higher HPV vaccine uptake in younger male participants than older ones suggests that the Swiss HPV vaccine recommendation for boys in 2015 led to a higher HPV vaccine uptake among younger eligible men. The finding of lower HPV vaccine uptake among older male youth point suggests a need for public health campaigns designed to increase awareness of catch-up vaccinations for male youth.

The VH prevalence in our study was almost 10 percentage points lower in men than in women, but this difference might be influenced by the fact that female participants were selected in a less random sample compared with men, who mostly were recruited through military enlistment. Still, this finding is consistent with previous reports suggesting that, in general, women tend to be more VH than men.[46] However, as suggested by the WHO's Strategic Advisory Group of Experts on Immunization working group on VH, a high level of VH leads to a lower vaccine demand, but lower VH levels do not necessarily lead to higher vaccine demand.[18]

The finding that VH is not significantly associated with HPV underimmunisation in men suggests that male youth face additional barriers to HPV vaccination, which may include limited HPV vaccine awareness and knowledge, access barriers and a lack of priority given to male HPV vaccination by public health authorities and providers. An explanation for higher access to the HPV vaccine among female youth is that catch-up HPV vaccination in young Swiss women is partially done by gynaecologists.[35] In contrast, young men in good health see a medical provider less often than young women in good health[47] and may, therefore, not receive an HPV vaccine

**Table 2** Unadjusted and adjusted odds of not having received ≥1 dose of HPV vaccine, controlling for sex, age groups, place of birth, language region, recruitment setting, availability of HPV vaccine school programme in the area of their residence and VH

| | Univariable analysis | Model 0 | Model 1 |
| --- | --- | --- | --- |
| | OR (CI 95%) | AOR (CI 95%) | AOR (CI 95%) |
| **Sex** | | | |
| Female | Reference | Reference | Reference |
| Male | 5.28 (3.17 to 8.80) | 5.16 (3.06 to 8.70) | 5.77 (3.40 to 9.79) |
| **Age groups based on birthdates** | | | |
| Born before 1 July 2002 | Reference | Reference | Reference |
| Born on or after 1 July 2002 | 0.74 (0.44 to 1.26) | 0.72 (0.42 to 1.21) | 0.7 (0.41 to 1.19) |
| **Place of birth** | | | |
| Outside Switzerland | Reference | Reference | Reference |
| Switzerland | 1.85 (0.95 to 3.60) | 2.03 (1.02 to 4.06) | 2.05 (1.02 to 4.10) |
| **Language region** | | | |
| German | Reference | Reference | Reference |
| French | 0.79 (0.33 to 1.89) | 0.89 (0.39 to 2.04) | 0.87 (0.38 to 1.98) |
| Italian | 0.54 (0.26 to 1.12) | 0.57 (0.29 to 1.13) | 0.6 (0.31 to 1.17) |
| **Participant recruitment setting** | | | |
| Biomedical provider's office | Reference | Reference | Reference |
| CAM provider's office | 2.46 (1.33 to 4.55) | 2.29 (1.21 to 4.32) | 2.09 (1.13 to 3.89) |
| Military enlistment | 7.29 (2.64 to 20.09) | 2.6 (0.99 to 6.80) | 2.79 (1.15 to 6.79) |
| **HPV vaccine school programme in area of residence** | | | |
| Not available | Reference | Reference | Reference |
| Available | 1.37 (0.85 to 2.21) | 1.26 (0.72 to 2.19) | 1.2 (0.69 to 2.08) |
| **YAV-5 Score** | | | |
| Score<50 (not hesitant) | Reference | | Reference |
| Score≥50 (vaccine hesitant) | 2.75 (1.69 to 4.48) | | 3.2 (1.94 to 5.26) |

AOR, adjusted OR; CAM, complementary and alternative medicine; HPV, human papillomavirus; OR, Odds ratio; VH, vaccine hesitancy; YAV-5, Youth Attitudes about Vaccines Scale.

recommendation from any provider,[4 5] nor be aware of the benefits of HPV vaccine.[4 5 11–14]

To our knowledge, this is the first study that measures VH in youth using the YAV-5 instrument, an adapted version of the PACV survey, in order to examine associations between VH and HPV vaccine uptake. Similarly, we here show a significant association of youth VH and HPV underimmunisation in female youth using the YAV-5-score. Our results echo the findings of Preston and colleagues who identified an association between negative attitudes towards HPV vaccine and HPV underimmunisation among women, but not among male college students in Florida, USA.[24] We also found that women recruited in the offices of CAM providers have higher odds of being unvaccinated with HPV vaccine, even when controlling for youth VH. This is consistent with previous research showing that CAM usage is associated with underimmunisation.[48 49]

**Strengths and limitations**
Strengths of our research include this being the first study using the YAV-5 to measure VH in youth and to associate

VH with HPV vaccination uptake. Additional strengths of our study include integration of a large number of male youth from all three language regions of Switzerland, which allowed us to gain important insight into males' perspectives and vaccine uptake. Little research has been done on males' HPV vaccine attitudes and uptake, particularly in the Swiss setting.

Our findings are likely comparable to other high-income countries in which HPV vaccination programmes have recently been extended to young men and where HPV vaccine is covered by health insurance. Since the YAV-5 survey instrument has been previously validated to measure VH in youth,[43] our study results can lead to meaningful comparisons with other populations.

Our study also has limitations. There was no evidence in our study that the availability of an HPV school vaccination programme was associated with increased HPV vaccination uptake. This stands in contrast to a 2018 Swiss study on the association between school vaccination programmes and HPV vaccine uptake in young women.[2]

**Table 3** Unadjusted and adjusted odds of not having received ≥1 dose of HPV vaccine stratified by sex; controlling for participant sex, age groups, place of birth, language region, recruitment setting, availability of HPV vaccine school programme in the area of their residence and VH

| | Univariable analysis, male only (n=415) | Model 0, male only (n=415) | Model 1, male only (n=415) | Univariable analysis, female only (n=259) | Model 0, female only (n=259) | Model 1, female only (n=259) |
| --- | --- | --- | --- | --- | --- | --- |
| | OR (95% CI) | AOR (95% CI) | AOR (95% CI) | OR (95% CI) | AOR (95% CI) | AOR (95% CI) |
| **Place of birth** | | | | | | |
| Outside Switzerland | Reference | Reference | Reference | Reference | Reference | Reference |
| Switzerland | 2.1 (0.84 to 5.23) | 1.77 (0.69 to 4.51) | 1.71 (0.67 to 4.34) | 2.14 (0.66 to 6.95) | 2.1 (0.64 to 6.86) | 2.38 (0.68 to 8.32) |
| **Age groups based on birthdates** | | | | | | |
| Born before 1 July 2002 | Reference | Reference | Reference | Reference | Reference | Reference |
| Born on or after 1 July 2002 | 0.29 (0.12 to 0.71) | 0.34 (0.14 to 0.81) | 0.31 (0.13 to 0.73) | 0.94 (0.46 to 1.93) | 0.97 (0.48 to 1.97) | 1.14 (0.56 to 2.33) |
| **Language region** | | | | | | |
| German | Reference | Reference | Reference | Reference | Reference | Reference |
| French | 0.5 (0.11 to 2.33) | 0.72 (0.16 to 3.20) | 0.69 (0.16 to 3.03) | 0.88 (0.31 to 2.52) | 0.99 (0.34 to 2.90) | 0.94 (0.32 to 2.80) |
| Italian | 0.38 (0.12 to 1.21) | 0.57 (0.18 to 1.82) | 0.57 (0.18 to 1.80) | 0.44 (0.17 to 1.12) | 0.56 (0.20 to 1.58) | 0.59 (0.22 to 1.62) |
| **Participant recruitment setting** | | | | | | |
| Biomedical provider | Reference | Reference | Reference | Reference | Reference | Reference |
| CAM provider | 2.63 (0.66 to 10.45) | 1.59 (0.44 to 5.82) | 1.43 (0.39 to 5.20) | 3.08 (1.33 to 7.14) | 2.75 (1.19 to 6.33) | 2.52 (1.13 to 5.63) |
| Military enlistment | 2.49 (0.44 to 13.99) | 1.83 (0.52 to 6.47) | 1.84 (0.57 to 5.95) | 0.58 (0.03 to 11.81) | 0.55 (0.03 to 9.53) | 0.59 (0.03 to 10.29) |
| **HPV vaccine school programme in area of residence** | | | | | | |
| Not available | Reference | Reference | Reference | Reference | Reference | Reference |
| Available | 1.39 (0.67 to 2.91) | 1.21 (0.53 to 2.76) | 1.2 (0.53 to 2.72) | 1.74 (0.88 to 3.43) | 1.14 (0.48 to 2.71) | 1 (0.42 to 2.37) |
| **YAV-5 Score** | | | | | | |
| Score<50 (not hesitant) | Reference | | Reference | Reference | | Reference |
| Score≥50 (vaccine hesitant) | 1.68 (0.75 to 3.76) | | 1.9 (0.84 to 4.31) | 5.29 (2.71 to 10.33) | | 4.9 (2.53 to 9.50) |

AOR, adjusted OR ; CAM, complementary and alternative medicine; HPV, human papillomavirus; OR, odds ratio; VH, vaccine hesitant; YAV-5, Youth Attitudes about Vaccines Scale.

However, our study has important limitations for investigating the impact of school programmes due to the large age range of participants, a non-representative sample, lack of data on schooling received and the possibility that youth no longer live in the same area as when they would have received such schooling.

Our sample is not representative for Switzerland. Additional potential sources of bias result from us not being able to include participants who do not go to a physicians' office. This bias applies more to women in our study and less to men because the majority of men were recruited during military enlistment. Finally, VH persons might be less likely than non-VH persons to participate in a study examining VH . Future research could address these gaps by using a random sampling approach throughout the population with the aims of gathering representative data.

## CONCLUSION

By administering the YAV-5, a validated instrument for measuring VH in youth,[43] to a large sample of youth in Switzerland, our results suggest that VH may be a major determinant of HPV underimmunisation for female youth. While male youth had higher odds of being unvaccinated, VH was not associated with HPV underimmunisation in male youth. This suggests that issues other than VH, including HPV vaccine awareness and knowledge and access to HPV vaccination for youth who do not have regular contacts with the healthcare system, may still be predominant reasons for low vaccine uptake among male youth in Switzerland. This should be explored in further research. To increase HPV vaccine uptake in Switzerland among both male and female youth, improving the design of HPV vaccination programmes is essential. Future vaccination programmes should address both female and male youth by not only increasing awareness and knowledge around HPV but also by optimising access to HPV vaccines. This is a key issue for many eligible and generally healthy youth who do not have regular contacts with the healthcare system. Such efforts could be implemented through strengthening HPV vaccination in school programmes for adolescents and by efforts aimed at raising awareness and improving access for catch-up vaccinations.

**Author affiliations**
[1]University Department of Medicine, Kantonsspital Baselland, Bruderholz, Switzerland
[2]University of Basel, Basel, Switzerland
[3]Swiss Tropical and Public Health Institute, Basel, Switzerland
[4]Rekrutierungszentrum Aarau, Schweizer Armee, Aarau, Switzerland
[5]Rekrutierungszentrum Monte Ceneri, Schweizer Armee, Monte Ceneri, Switzerland

⁶Kompetenzzentrum für Militär- und Katastrophenmedizin, Eidgenössisches Departement für Verteidigung, Bevölkerungsschutz und Sport VBS Schweizer Armee, Ittigen, Switzerland
⁷Department of Pediatrics, HFR Fribourg Cantonal Hospital, Fribourg, Switzerland

**Acknowledgements** We would like to thank the participating adolescents, young adults and physicians and their medical assistants for their time and effort.

**Contributors** LMK and CLS recruited participating youth, conducted interviews, carried out the analysis, drafted the initial manuscript, and reviewed and revised the manuscript. KJ conceptualised and designed the study, designed the data collection instruments, coordinated and supervised data collection, carried out the analyses and reviewed and revised the manuscript. AM and NR recruited participating youth, conducted interviews and reviewed and revised the manuscript. SSM, FM and NG recruited participating youth, and reviewed and revised the manuscript. BMH conceptualised and designed the study, recruited participating providers and reviewed and revised the manuscript. SM designed the data collection instruments, coordinated and supervised data collection and reviewed and revised the manuscript. MJD conceptualised and designed the study, designed the data collection instruments and reviewed and revised the manuscript. PET conceptualised and designed the study, designed the data collection instruments, recruited participating providers, coordinated and supervised data collection, carried out the analyses, and reviewed and revised the manuscript. PET is the guarantor: he accepts full responsibility for the work and/or the conduct of the study, had access to the data, and controlled the decision to publish. All authors approved the final manuscript as submitted and agree to be accountable for all aspects of the work. All authors attest they meet the ICMJE criteria for authorship.

**Funding** The study was funded via the Swiss National Science Foundation (Grant Number 407440_167398, recipient: PET), in the setting of National Research Programme (NRP) 74. The study benefited from supplementary postdoctoral fellowship funding from the Nora van Meeuwen-Haefliger-Foundation. No funding was obtained from vaccine manufacturers or the Swiss Federal Office of Public Health.

**Competing interests** None declared.

**Patient and public involvement** Patients and/or the public were not involved in the design, or conduct, or reporting, or dissemination plans of this research.

**Patient consent for publication** Not applicable.

**Ethics approval** This study involves human participants and was approved by Ethikkommission Nordwest- und Zentralschweiz, EKNZ; project ID number 2017-00725. Participants gave informed consent to participate in the study before taking part.

**Provenance and peer review** Not commissioned; externally peer reviewed.

**Data availability statement** Data are available upon reasonable request. Data are available upon reasonable request to the corresponding author.

**ORCID iDs**
Laura M Kiener http://orcid.org/0000-0002-7505-3671
Corina L Schwendener http://orcid.org/0000-0002-0315-7949
Michael J Deml http://orcid.org/0000-0003-2224-8173
Philip E Tarr http://orcid.org/0000-0002-1488-5407

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
