## [Reviewer comments · BMJ Open]

ARTICLE DETAILS

TITLE (PROVISIONAL)	Vaccine hesitancy and HPV vaccine uptake among male and female youth in Switzerland: a cross-sectional study
AUTHORS	Tarr, Philip; Kiener, Laura M.; Schwendener, Corina L.; Jafflin, Kristen; Meier, Audrey; Reber, Noah; Schärli Maurer, Susanna; Muggli, Franco; Gültekin, Nejla; Huber, Benedikt; Merten, Sonja; Deml, Michael

VERSION 1 – REVIEW

REVIEWER	Bednarczyk, Robert A. Emory University, Hubert Department of Global Health
REVIEW RETURNED	14-Sep-2021

GENERAL COMMENTS	Suboptimal HPV vaccination remains an intractable problem. This study evaluates a new adolescent/young adult vaccine hesitance scale with regard to HPV vaccination in Switzerland. While conceptually this manuscript reports a fairly well-designed study, there are numerous issues that reduce my enthusiasm. 1. Overall, the age issues, both for differences in timing for males and females with regard to hpv vaccine recommendations and sampling, are well-described, but the overall analysis and reporting seems to lose some of the context around this. For example, there were lower odds of being unvaccinated among younger males, but the manuscript does not sufficiently address the issue of higher demand after the recommendation was made for males versus the need to get older males in for catch-up vaccination. That difference can be a major impact on demand.2. The ages of hpv vaccine recommendation in Switzerland are presented in the methods, but when discussing the recommendations in the introduction, it would be helpful to have that information presentation.3. In the last sentence of the introduction, when presenting the hypotheses, it is unclear where the hypothesis of stronger association between hesitance score and vaccination among males would come from. In the discussion, it is mentioned that there is regularly more hesitance identified in females, but no indication of how this would specifically translate to a stronger association in females.4. The cutpoint of 50 for vaccine hesitant/not hesitant is unclear - did this come from the validation study that is being written up currently? Additionally, was there any robustness testing (for example, does someone with a 48-52 score look more like they are hesitant or not, based on vaccine uptake)?5. Vaccine records were queried - I have a question based on my lack of knowledge of the Swiss health system - is there any
--

	national health or immunization registry? Or did this rely on individual vaccine cards? 6. Statistical issues: 6a. Why was odds ratio calculated, especially with relatively high vaccine uptake levels? For example, where the OR for unvaccination was in the range of about 5, computing out a prevalence ratio gives a value of about 2 - still significant (both statistically and clinically), but at a very different magnitude. 6b. why was there no assessment of effect modification by age, given the different sampling strategies and age distributions, especially for males where there historically have been differences in vaccination recommendation by age? 6c. Was there any assessment of frequency of care, since this is described in the fourth paragraph of the strengths and limitations as potential source of bias? 6d . The proportions for language region for the total sample in Table 1 seem reversed for Italian and French.
--	---

REVIEWER	Reno, Jenna University of Colorado Denver, Family Medicine
REVIEW RETURNED	12-Oct-2021

GENERAL COMMENTS	Overall, this is a strong study that addresses an important yet understudied topic -- the role of vaccine hesitancy among adolescents and older youth as it pertains to vaccine uptake. The manuscript would be strengthened by making the following additions: 1) Provide explicit justification for the second hypothesis in the Introduction. 2) The objective of the study is stated as "Identifying factors associated with HPV vaccine uptake is essential for designing successful vaccination programs." However, the discussion is limited in its discussion of explicit implications for the design of vaccination programs based on the results of this study.
---

VERSION 1 – AUTHOR RESPONSE

Reviewer #1: Dr. Robert A. Bednarczyk, Emory University

(....)

1.) Overall, the age issues, both for differences in timing for males and females with regard to hpv vaccine recommendations and sampling, are well-described, but the overall analysis and reporting seems to lose some of the context around this. For example, there were lower odds of being unvaccinated among younger males, but the manuscript does not sufficiently address the issue of higher demand after the recommendation was made for males versus the need to get older males in for catch-up vaccination. That difference can be a major impact on demand.

Response: In the second paragraph of the discussion we already point out the impact of the new HPV vaccine recommendation for male youth in 2015 which led to a higher HPV vaccine uptake in younger males. We have changed the second part of the conclusion emphasizing the importance of HPV vaccine awareness in catch up vaccinations also for older male youth through public health authorities as following:

“To increase HPV vaccine uptake in Switzerland among both male and female youth, improving the design of HPV vaccination programs is essential. Future vaccination programs should address both female and male youth by not only increasing awareness and knowledge around HPV but also by optimizing access to HPV vaccines. This is a key issue for many eligible and generally healthy youth who do not have regular contacts with the healthcare system. Such efforts could be implemented through strengthening HPV vaccination in school programs for adolescents and by efforts aimed at raising awareness and improving access for catch-up vaccinations.”

2.) The ages of hpv vaccine recommendation in Switzerland are presented in the methods, but when discussing the recommendations in the introduction, it would be helpful to have that information presentation.

Response: The fourth part of the introduction says that the HPV vaccine recommendation has been made by the FOPH for females in 2007 and for males as optional recommendation in 2015. We have added the age groups for the routine vaccinations and catch-up vaccinations to the introduction:

“The Swiss Federal Office of Public Health (FOPH) and the Federal Vaccination Commission have recommended HPV vaccine since 2007 as a routine vaccine for girls 11 to 14 years of age and young women as catch-up vaccination 15 to 26 years of age. In 2015, the FOPH added HPV vaccine to the list of optional vaccines for boys and young men with the same age groups as for females.”

3.) In the last sentence of the introduction, when presenting the hypotheses, it is unclear where the hypothesis of stronger association between hesitance score and vaccination among males would come from. In the discussion, it is mentioned that there is regularly more hesitance identified in females, but no indication of how this would specifically translate to a stronger association in females

Response: We expect a weaker association between VH and HPV vaccine uptake in males because of its later introduction and therefore lower awareness of the HPV vaccine. We expect other factors than VH such as awareness and access barriers to more strongly influence HPV vaccine uptake in males compared to the association between VH and HPV vaccine uptake in females. We added an explanation about this in the introduction, last paragraph:

“With the later introduction of the HPV vaccine recommendation for males, we expect factors other than VH, such as HPV vaccine awareness, to influence uptake in males.”

4.) The cutpoint of 50 for vaccine hesitant/not hesitant is unclear - did this come from the validation study that is being written up currently? Additionally, was there any robustness testing (for example, does someone with a 48-52 score look more like they are hesitant or not, based on vaccine uptake?)

Response: The cutpoint of 50 comes from the original PACV survey which was designed and validated by Opel and colleagues (our references 20 and 38). We use it based on past usage: the cutpoint of 50 is fairly standard in the literature. As we modified this tool to the YAV-5 (Youth attitudes about Vaccines Scale) we also validated it with exploratory factor analysis and Mokken scale analysis (our reference 43). However, we did not test robustness the way the reviewer is suggesting.

5. Vaccine records were queried - I have a question based on my lack of knowledge of the Swiss health system - is there any national health or immunization registry? Or did this rely on individual vaccine cards?

Response: In Switzerland, there is no national immunization registry. Instead, those seeking vaccination services receive individual vaccine booklets (records) on which the healthcare provider

documents the administered vaccines. Therefore, the information presented in this study rely on the vaccine certificates provided by study participants, as is already pointed out in the methods, section 2.3.

6. Statistical issues:

6a. Why was odds ratio calculated, especially with relatively high vaccine uptake levels? For example, where the OR for unvaccination was in the range of about 5, computing out a prevalence ratio gives a value of about 2 - still significant (both statistically and clinically), but at a very different magnitude.

Response:

"We agree that odds ratios are significantly larger than risk ratios (prevalence ratios) in our case. However, it is the standard practice in the literature, and allows easier adjustment, as here, with multi-level models with random effects to account for clustering. As such, we chose to maintain use of this measure, clearly stating that we calculate odds ratios and discussing our findings using appropriate terms. However, to add clarity, we have therefore added the following sentence to the methods section, 2.7 Quantitative data analysis:

"Due to the high prevalence of vaccine hesitancy, calculated odds ratios are significantly higher than prevalence ratios would be, however, as these models are standard in the literature and allow easier adjustment for sampling frames, we use them here."

6b. why was there no assessment of effect modification by age, given the different sampling strategies and age distributions, especially for males where there historically have been differences in vaccination recommendation by age?

Response: Based on this feedback, we tested whether there was an interaction effect between age, using our binary age measure, and vaccination hesitancy. There was no significant association of the interaction term and vaccine uptake in the full model or in male- and female-only models. Results available on request. We have therefore added the following sentences to the methods, section 2.7:

"Additionally we also checked for an interaction between vaccine hesitancy and age, but found none. Results of those analyses are available on request."

6c. Was there any assessment of frequency of care, since this is described in the fourth paragraph of the strengths and limitations as potential source of bias?

Response: No, this was not assessed since we did not collect information on youth's frequency of care. However, we see the frequency of care as a potential bias in our sampling strategy as only female patients who regularly visit a health care provider could get included in this study. Since males were mainly recruited through the military enlistment, we expect frequency of care as a less strong bias for males. We have therefore added the following sentence to the discussion, section "strengths and limitations":

„This bias applies more to females in our study and less to males because the majority of males were recruited during military enlistment.“

6d . The proportions for language region for the total sample in Table 1 seem reversed for Italian and French.

Response: Due to our sampling strategy the Italian speaking language region is slightly overrepresented. For a better overview we have now modified table 1 and have switched the order of the Italian and French language region, the proportions for language regions are now in the correct

way.

Reviewer #1: Dr. Jenna Reno, University of Colorado Denver

(....)

1.) Provide explicit justification for the second hypothesis in the Introduction.

Response: We expect a weaker association between VH and HPV vaccine uptake in males than in females because of its later introduction and therefore lower awareness of the HPV vaccine. Otherwise stated, in males, we expect factors other than VH such as awareness and access barriers to influence HPV vaccine uptake. We added an explanation about this in the introduction:

“With the later introduction of the HPV vaccine recommendation for males, we expect factors other than VH, such as HPV vaccine awareness, to influence uptake in males.”

2.) The objective of the study is stated as "Identifying factors associated with HPV vaccine uptake is essential for designing successful vaccination programs." However, the discussion is limited in its discussion of explicit implications for the design of vaccination programs based on the results of this study.

Response: We have added explicit suggestions in the conclusion section for future improvement of vaccination programs in Switzerland. Therefore, we think it is essential to address male and female youth, not only to increase awareness and knowledge about HPV vaccine, but also to make the vaccine more accessible, eg. through school programs but also through improved catch-up vaccination programs .

“To increase HPV vaccine uptake in Switzerland among both male and female youth, improving the design of HPV vaccination programming is essential. Future vaccination programs should address both female and male youth by not only increasing awareness and knowledge around HPV but also by optimizing access to HPV vaccines. This is a key issue for many eligible and generally healthy youth who do not have regular contacts with the healthcare system. Such efforts could be implemented through strengthening HPV vaccination in school programs for adolescents and by efforts aimed at raising awareness and improving access for catch-up vaccinations.”

VERSION 2 – REVIEW

REVIEWER	Bednarczyk, Robert A. Emory University, Hubert Department of Global Health
REVIEW RETURNED	11-Feb-2022
GENERAL COMMENTS	All prior comments have been sufficiently addressed. I appreciate the additional analyses conducted, and the clarifications provided.